# Preservation of collagen in the soft tissues of frozen mammoths

**Shunji Hattori** [1]*, **Tomomi Kiriyama-Tanaka**[1], **Masashi Kusubata**[1], **Yuki Taga**[1],
**Testuya Ebihara**[1], **Yuki Kumazawa**[1], **Katsuyuki Imai**[2], **Mitsutaka Miura**[2],
**Yoshihiro Mezaki** [2], **Alexei Tikhonov**[3], **Haruki Senoo**[2]

1 Nippi Research Institute of Biomatrix, Toride, Ibaraki, Japan, 2 Department of Cell Biology and Histology,
Akita University School of Medicine, Akita, Japan, 3 Zoological Institute, Russian Academy of Sciences,
St. Petersburg, Russia

* shunhatt.ten@icloud.com

## Abstract

We investigated the characteristics of extracellular matrix (ECM) in the soft tissue of two frozen baby woolly mammoths (*Mammuthus primigenius*) that died and were buried in Siberian permafrost approximately 40,000 years ago. Morphological and biochemical analyses of mammoth lung and liver demonstrated that those soft tissues were preserved at the gross anatomical and histological levels. The ultrastructure of ECM components, namely a fibrillar structure with a collagen-characteristic pattern of cross-striation, was clearly visible with transmission and scanning electron microscopy. Type I and type IV collagens were detected by immunohistochemical observation. Quantitative amino acid analysis of liver and lung tissues of the baby mammoths indicated that collagenous protein is selectively preserved in these tissues as a main protein. Type I and type III collagens were detected as major components by means of liquid chromatography–mass spectrometry analysis after digestion with trypsin. These results indicate that the triple helical collagen molecule, which is resistant to proteinase digestion, has been preserved in the soft tissues of these frozen mammoths for 40,000 years.

**Data Availability Statement:** We have deposited the MS datasets to the ProteomeXchange consortium via jPOST partner repository with the dataset identifier PXD024006 (https://repository.jpostdb.org/entry/JPST001078.0).

## Introduction

Collagen is an extracellular matrix protein and occurs in all types of animal tissue. The triple helix structure of collagen is very stable and is resistant to attack by most proteinases other than special collagenolytic enzymes [1]. Collagen is greatly affected by post-translational modifications, such as hydroxylation of proline and lysine, cross-link formation, and glycosylation of lysine, that are not directly defined by DNA information [2]. For this reason, collagen analysis can yield unique information about extinct animals.

Several studies have detected collagen in hard tissue (e.g. bone and dentine) from mammoths (*Mammuthus columbi*) [3–5], but there have been no reports on protein analysis of mammoth soft tissue. Here we analysed soft tissues from two frozen baby mammoths, nicknamed Masha (female) and Dima (male), that were excavated in Siberia and are preserved at the Russian Academy of Sciences (St. Petersburg) [6]. We analysed tissue from these baby

**Funding:** The author(s) received no specific funding for this work.

**Competing interests:** The authors have declared that no competing interests exist.

**Abbreviations:** ECM, extracellular matrix; PBS, phosphate-buffered saline; DSC, differential scanning calorimetry); LC–MS, liquid chromatography–mass spectrometry.

mammoths using histochemical and biochemical methods, and found that collagen was preserved well in the tissue.

DNA sequence information is a powerful tool for evolutionary research on extinct animals, including mammoths [7]. Our results suggest that collagen can be used for protein research on extinct animals.

## Materials and methods

### Mammoth and porcine tissues

Liver, lung and genitals tissue were obtained from two frozen baby woolly mammoths (*Mammuthus primigenius*, named Dima [8] and Masha [9]) that died and were buried in Siberian permafrost approximately 40,000 years ago. The tissues were kept in an ethanol solution at the Russian Academy of Sciences. Bone (dried) and kidney (fixed in formaldehyde) of an Indian elephant (*Elephas maximus indicus*) were given to us by Dr. Yasuhiro Kon (Hokkaido University). Fresh porcine tissues (liver, lung and skin) and bovine skins were purchased from a commercial meat supplier (Tokyo Shibaura Zoki, Japan). Porcine and bovine was slaughtered according to Japanese slaughterhouse act. No permits were required for the described study, which compiled with all relevant regulations.

### Light microscopy

Tissue sections were stained with haematoxylin and eosin, and connective tissue was detected by azan staining or Ishii–Ishii's silver impregnation [10] as described previously [11, 12]. For haematoxylin and eosin staining, several blocks were immersed in 3.7% formaldehyde, dehydrated in a series of graded ethanol, and embedded in paraffin. For Ishii–Ishii's silver impregnation, several blocks were immersed in 3.7% formaldehyde and cut into 10-μm slices with a freezing microtome. Photographs were taken by Axiovision (Carl Zeiss, Oberkochen, Germany).

### Immunohistochemistry

Immunofluorescence staining of type I and type IV collagens was performed using cryosections. Tissues were embedded in OTC compound and cut at 10-μm thickness by cryostat (Leica biosystems, Nussloch, Germany). The cryosections were first incubated for at least 15 min with 10% skim milk in phosphate-buffered saline (PBS) to block nonspecific binding of antibodies, rinsed with PBS, and incubated overnight with rabbit anti-type I collagen antibody LB-1102 (LSL,Japan) diluted 1:500 or goat anti-type IV collagen antibody: sc-9301 (Santa Cruz Biotecnology,TX,USA) diluted 1:100 in 5% SM. The sections were then rinsed with PBS, incubated for 30 min with Alexa Flour 488 conjugated donkey anti-rabbit IgG (Abcam, Cambridge, UK) or Alexa Flour 488 conjugated donkey anti-goat IgG (Abcam, Cambridge, UK) diluted 1:100 in 5% SM, rinsed, and finally stained with 1 μM To-Pro-3(Invitrogen, CA, USA) for 30min. Sections were scanned by LSM 510 (Carl Zeiss, Oberkochen, Germany)

### Scanning electron microscopic observation

Samples were fixed for 4 h with Karnovsky's fixative solution (2.5% glutaraldehyde and 2% paraformaldehyde in phosphate buffer of pH 7.4). After washing with PBS, the specimens were fixed for 1 h with 1% osmium acid in 0.1 M phosphate buffer (pH 7.4), dehydrated with ethanol, and dried in an HCP-2 critical point dryer (Hitachi, Tokyo, Japan). The dried specimens were fixed on specimen stages with double-sided adhesive tape and then coated with platinum

in a sputter coater (Model 108 Auto, Cressington Co., Watford, UK). Specimens were observed under an S-4500 electron microscope (Hitachi).

## Transmission electron microscopy

A block from each organ was perfused with 1.5% glutaraldehyde in 0.062 M cacodylate buffer, pH 7.4, containing 1% sucrose for 1 or 2 min by injection of the block. After perfusion, the tissue blocks were prefixed in 2% osmium tetroxide in 0.1 M phosphate buffer (pH 7.4) for 2 h at 4°C, dehydrated in a graded ethanol series, and embedded in Epon 812. Ultrathin sections were made with an ultramicrotome 2088-V (LKB, Stockholm, Sweden) and stained with 2% uranyl acetate. The sections were examined with a transmission electron microscope 1200EX (JEOL, Tokyo, Japan) at an acceleration voltage of 80 kV [13]. Thick sections were examined under a light microscope after staining with 1% toluidine blue containing 1% borax.

## Amino acid analysis and quantification of collagen in the tissues

Mammoth tissues (Masha) and fresh porcine tissues were hydrolyzed in 6 M HCl at 110°C for 24 h. After evaporation of HCl, the sample was resuspended in water and analysed with an L-8800 amino acid analyser (Hitachi). The collagen content in each tissue was evaluated based on the amount of hydroxyproline (a specific amino acid in collagen) from the amino acid analysis data. The hydroxyproline weight and total amino acid content in each tissue were calculated based on the amino acid composition of porcine type I collagen as a standard [14]. Total protein weight was estimated from the total amount of amino acids. We conducted 3 times (mammoth liver and lung) or 4 times (mammoth genitals, porcine liver, lung and skin) independent analyses using tissue samples from same animal.

## Differential scanning calorimetry analysis

Differential scanning calorimetry (DSC) analysis was carried out on mammoth lung and bovine dermis. Samples dipped in PBS were sealed in an aluminum pan and placed in a Thermal Analysis System Perkin-Elmer 7 series (Perkin-Elmer, MA, USA). The temperature was increased from 30°C to 90°C at a rate of 2°C/min in nitrogen gas flow, and the heat flow was monitored [15]. Indium (melting point 156.4°C) was used for temperature correction.

## Liquid chromatography–mass spectrometry analysis

Lung and liver samples from the mammoths and the kidney sample from the Indian elephant were washed with $NH_4HCO_3$ and homogenized using a BioMasher (Nippi, Tokyo, Japan). The bone sample from the Indian elephant was decalcified with EDTA. After heating at 80°C for 5 min, the samples were digested with trypsin (1 mg/mL) (Sigma-Aldrich,MO,USA) at 37°C overnight. Trypsin digestion was repeated at 37°C for 6 h. The digested sample (supernatant) was desalted by Sep-Pak C18 (Waters, Milford, MA, USA). The sample was analysed by liquid chromatography–mass spectrometry (LC–MS) using a 3200 QTRAP hybrid triple quadrupole/linear ion trap mass spectrometer (AB Sciex, Foster City, CA, USA) coupled with an Agilent 1200 series high-performance liquid chromatography system (Agilent Technologies, Palo Alto, CA, USA). Tryptic peptides were separated using an XBridge C18 column (3.5 μm particle size, 150 mm length × 2.1 mm inner diameter, Waters) with an acetonitrile gradient, and MS/MS spectra were acquired by selecting the two most intense precursor ions of the prior survey MS scan. The acquired MS/MS spectra were searched against the UniProtKB/Swiss-Prot database (release 2018_05) using ProteinPilot software 4.0 (AB Sciex). Search parameters included digestion by trypsin, biological modifications ID focus, amino acid

substitutions ID focus, and 95% protein confidence threshold. Search criteria of post-translational modifications were optimized for collagen analysis as described previously [16]. We have deposited the MS datasets to the ProteomeXchange consortium via the jPOST partner repository with the dataset identifier PXD024006 (https://repository.jpostdb.org/entry/JPST001078.0).

### Collagen extraction by pepsin digestion and alkali treatment

Each tissue sample (100 mg) was washed with PBS, suspended in acetic acid (0.5 M) and homogenized. Pepsin (Sigma-Aldrich,MO,USA) (final concentration 1 mg/mL) was added to the homogenized sample, which was incubated for 24 h at 4˚C. After centrifugation, the supernatant was used as the pepsin fraction. The precipitate was suspended in alkali solution (0.075 N NaOH and 0.06 M monomethylamine) for 7 days at 4˚C. After centrifugation, the supernatant was used as the alkali fraction. The extracted fractions were subjected to SDS-PAGE analysis followed by Coomassie or silver staining [17, 18].

## Results and discussion

### Histological observation of mammoth tissue

The lung, liver and genitals of Masha were preserved at the gross anatomical level (Fig 1A). The bone and kidney samples of Indian elephant were also shown (Fig 1B). Histological observation of Dima's liver showed preserved extracellular structure (Fig 2A), and collagen fibers were detected by Ishii–Ishii's silver plating and azan staining (Fig 2B and 2C). Scanning electron microscope images of liver (Fig 3A) and lung (Fig 3B) samples showed a fibrillar structure with a characteristic pattern of cross-striations and a basement-membrane-like structure. The lung specimen from Dima was examined by transmission electron microscopy (Fig 4A), and collagen-specific banding structures with a period of 67 nm were observed at high magnification (Fig 4B). Positive reactions to anti-type IV (Fig 5A) and type I (Fig 5B) collagen antibodies were observed in extracellular matrix (ECM) components of the lung specimen from Masha by immunofluorescence. These results indicate that ECM components, including collagens, were stable and have been preserved in the tissues of these frozen mammoths for 40,000 years.

### Amino acid analysis of tissues

The histological data suggested that a protein component is preserved in the frozen mammoth tissue; therefore, amino acid analysis of liver and lung whole tissue was conducted. Bovine lung, porcine liver and porcine skin fresh tissue were analysed as controls (Table 1). Mammoth lung and liver contained higher amounts of hydroxyproline than the corresponding bovine and porcine fresh tissues. The hydroxyproline content in mammoth tissue is equivalent to that of skin, which is rich in collagen.

The protein content per dry tissue weight was estimated by weighing of wet and dried tissue and amino acid analysis (Table 2). The protein content in mammoth tissue is lower than that in equivalent fresh tissue. Fresh lung, liver and skin contain 70% protein by weight of dried tissue (Fig 6A). For the mammoth samples, the protein content as a percentage of dry weight is 46% of lung tissue, 3.4% of liver tissue and 6.8% of genitals tissue (Fig 6A). The profile of the collagen content by weight of dried tissue in mammoth tissue is resemble to that of the protein content by weight dried tissue. On the other hand, the collagen content of fresh lung is smaller than that of skin. (Fig 6B). These results may indicate that much of the protein in the mammoth soft tissue has been degraded. This result may indicate that much of the protein in the mammoth soft tissue has been degraded. We also calculated the collagen ratio per total protein

a

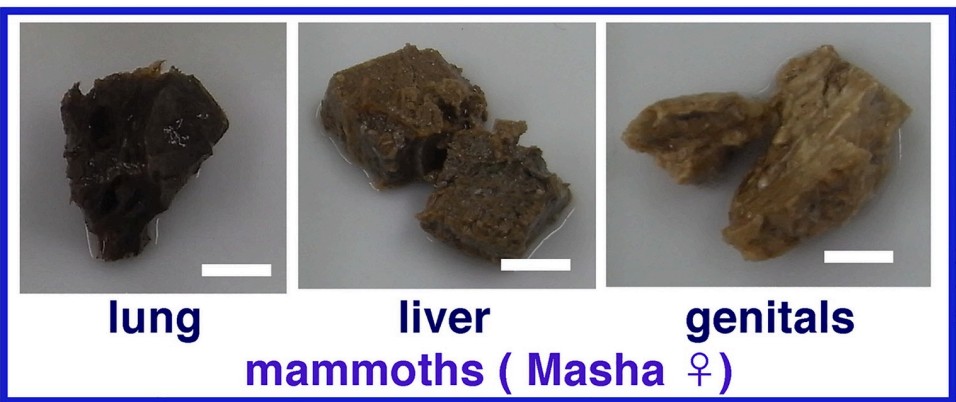

b

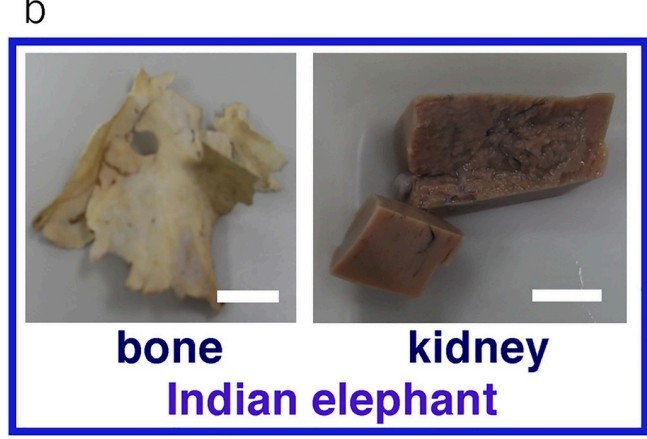

**Fig 1. Sample pieces from mammoth and elephant tissues. (a)** Sample pieces of lung, liver and genitals from baby mammoth (Masha) that were preserved in ethanol solution. Scale bar is 1 cm. **(b)** Sample pieces of bone and kidney from Indian elephant. Scale bar is 1cm.

of each tissue using the total amino acid and hydroxyproline content data (Fig 6C). The collagen ratio against total protein of mammoth lung, liver and genitals are 70%, 71% and 68% respectively; in contrast, the collagen content in total protein of fresh tissue samples from lung and liver is 20% and 14%, respectively. Collagen ratio in skin dermis is 84%. The collagen ratio in mammoth lung, liver and genitals are 70%, 71% and 68%. These are comparable with that of fresh skin dermis. These results suggest that most proteins other than collagen have been degraded during the long elapsed time, even in animals preserved in ice. In other words, collagen molecules are very stable with respect to weathering.

1. This data was presented at the 4th Pan-Pacific Connective Tissue Societies Symposium. November 15–19, 1999 (Queenstown, New Zealand) by Ebihara T. et al.

2. Amino acid analysis data of published mammoth bone data from Schaedler et al. [3] was provided as controls.

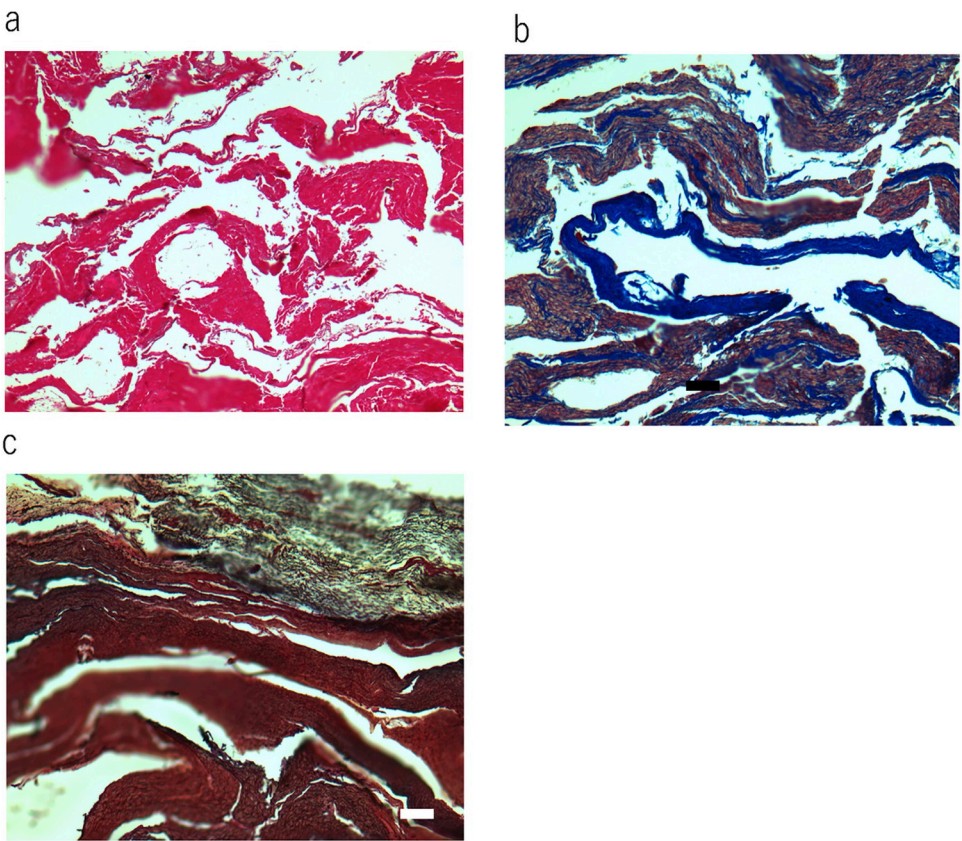

**Fig 2. Histochemical observation of soft tissue of mammoths.** (a) Haematoxylin and eosin staining of liver section from Dima. Scale bar is 100 μm. (b) Azan staining of liver section from Dima. Scale bar is 100 μm. (c) Ishii–Ishii's silver impregnation of liver section from Dima. Scale bar is 100 μm.

## DSC analysis of mammoth lung

Histochemical observation and amino acid data suggested that collagen fibrils were preserved in the mammoth soft tissue. Using DSC, we examined whether higher-order collagen protein structure was preserved. The melting endothermic behaviour of bovine dermis exhibited a

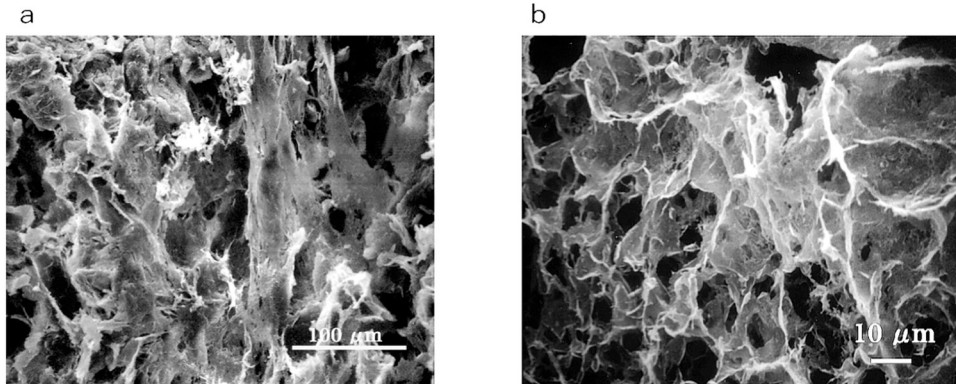

**Fig 3. Scanning electron microscope images of mammoth soft tissues.** (a) Liver section from Masha. (b) Lung section from Masha.

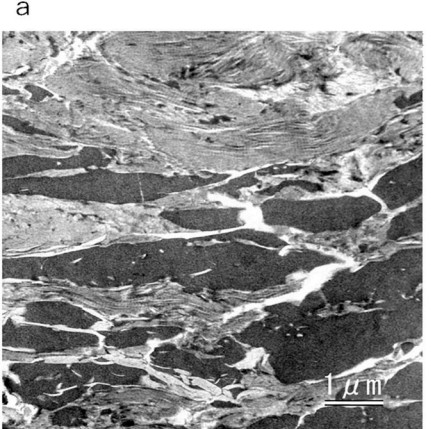
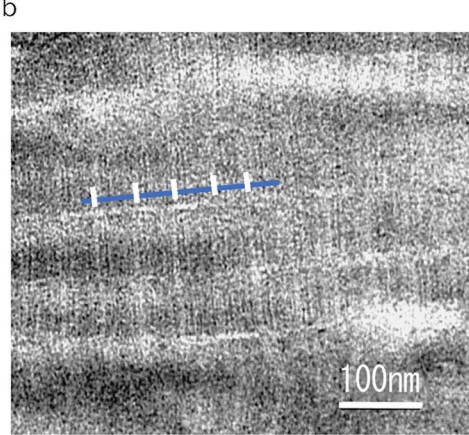

**Fig 4. Transmission electron microscope images of mammoth soft tissues.** (a) Lung section from Dima. (b) Higher magnification of same section. The blue bar indicates collagen molecule length; the white bars indicate the 67-nm-period banding pattern of collagen fibers.

peak endotherm temperature of 70°C (Fig 7), which is typical for collagen denaturation in skin. Mammoth lung tissue showed two peaks (74°C and 80°C) on the endothermy curve (Fig 7). The peak temperatures on the melting endothermic curves were 4–10°C higher for mammoth tissue than for bovine dermis; thus, some collagen-specific higher-order protein structure may be preserved in mammoth tissues. This difference may be attributed to differences in the nature of collagen fibrils or the different levels of cross-linkage in collagen chains that occurred during preservation in ice or in samples preserved in ethanol after excavation.

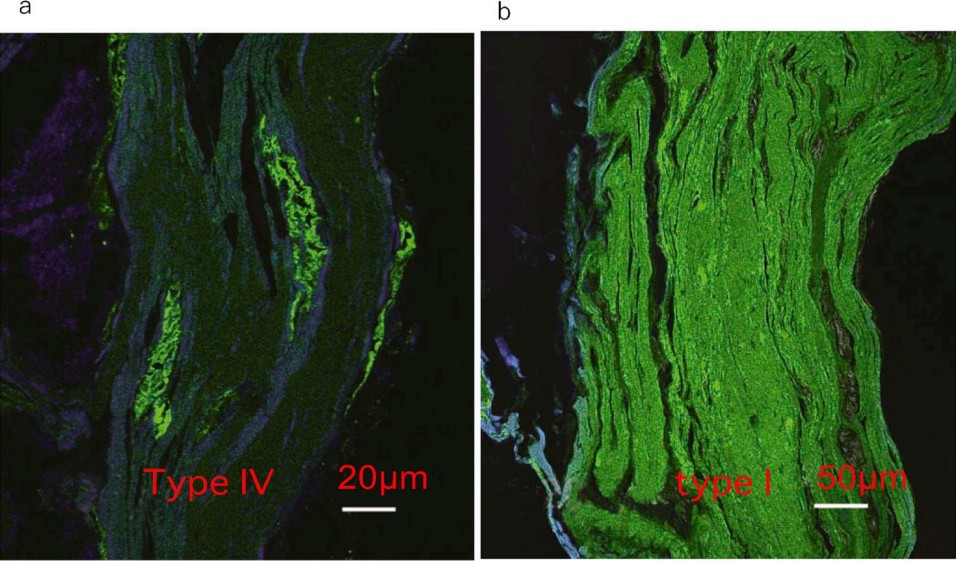

**Fig 5. Immunofluorescence observation of mammoth soft tissues.** (a) Masha's lung stained by anti-type IV collagen antibody. (b) Masha's lung stained by anti-type I collagen antibody.

**Table 1. Amino acid content (Residues/1000 residues) of mammoth (Masha) lung and liver, bovine lung, and porcine liver and skin.** Representative data out of three or four analyses.

|  | Mammoth (Masha) | | | Porcine | | | | mammoth |
|---|---|---|---|---|---|---|---|---|
|  | Lung | Liver | genitals | Lung | Liver | skin | purified col.I[1] | Bone[2] |
| Hyp | 69.2 | 68.4 | 64.8 | 17.5 | 9.6 | 73.2 | 93.6 | 95.3 |
| Asp | 62.3 | 68.3 | 77.7 | 86.3 | 85.3 | 51.5 | 42.1 | 44.9 |
| Thr | 30.8 | 33.1 | 35.7 | 45.9 | 47.5 | 20.4 | 16.1 | 18.9 |
| Ser | 43.0 | 44.4 | 42.3 | 55.2 | 54.3 | 41.0 | 32.3 | 36.2 |
| Glu | 83.1 | 82.0 | 88.8 | 108.6 | 111.1 | 84.1 | 72.2 | 72.3 |
| Pro | 94.1 | 88.6 | 86.0 | 63.1 | 56.7 | 111.9 | 131.2 | 124.2 |
| Gly | 264.7 | 259.2 | 222.6 | 141.1 | 103.6 | 307.4 | 336.3 | 337.2 |
| Ala | 97.6 | 100.2 | 104.5 | 93.9 | 89.7 | 105.2 | 118.7 | 97.00 |
| Cys | 4.3 | 0.0 | 7.4 | 8.6 | 4.5 | 2.6 | 0.0 | 1.1 |
| Val | 40.0 | 44.5 | 60.8 | 65.6 | 62.9 | 26.6 | 21.0 | 24.9 |
| Met | 6.2 | 0.4 | 6.8 | 11.0 | 20.6 | 7.8 | 5.6 | 6.1 |
| Ile | 25.8 | 18.3 | 21.1 | 35.6 | 44.8 | 14.7 | 8.9 | 10.1 |
| Leu | 62.6 | 61.9 | 68.3 | 81.3 | 94.1 | 33.7 | 22.7 | 26.1 |
| Tyr | 3.3 | 3.0 | 5.1 | 16.0 | 28.1 | 8.3 | 2.0 | 6.1 |
| Phe | 29.7 | 31.3 | 31.3 | 35.1 | 40.2 | 16.7 | 11.7 | 14.4 |
| Hyl | 9.2 | 11.1 | 5.4 | 3.3 | 3.5 | 0.5 | 7.0 | 3.2 |
| Lys | 15.5 | 20.6 | 21.8 | 61.5 | 72.4 | 31.5 | 26.1 | 28.7 |
| His | 7.9 | 17.1 | 4.7 | 21.6 | 22.7 | 7.7 | 4.0 | 2.5 |
| Arg | 48.8 | 45.3 | 44.8 | 48.8 | 43.5 | 48.9 | 48.5 | 47.5 |

Hyp;Hydroxyproline, Hyl; Hydroxylysine.

## LC–MS analysis of tissues

After digestion of Masha's lung tissue with trypsin, type I α1, type α2, type III collagen were detected as major components by LC–MS identification using publicly available sequences from other species (Table 3). In addition, type XXIV collagen was identified as a minor component. Hemoglobin β chain was also found, which may be from blood. We could not detect type IV collagen by this method. We conducted the same analysis on Masha's liver and detected type I α1, type α2 and type III collagen. We also analysed the Indian elephant bone and kidney samples in the same way as controls, and detected collagen of types I α1 and α2 in the bone sample and of type I α2 and type IV α4 in the kidney sample (Table 3). Hence, collagen is preserved well in frozen mammoth soft tissues.

**Table 2. Estimated protein and collagen content in soft tissues from the amino acid analysis data.** Representative data out of three or four analyses.

|  | Mammoth | | | Porcine | | |
|---|---|---|---|---|---|---|
| sample | lung | liver | genitals | lung | liver | skin |
| Wet Weight (mg) | 40.0 | 89.0 | 51.2 | 63.6 | 94.0 | 80.0 |
| Dry Weight (mg) | 15.3 | 35.7 | 18.5 | 12.5 | 29.0 | 21.5 |
| Protein (mg) | 6.3 | 0.9 | 1.24 | 10.0 | 12.1 | 15.8 |
| Collagen (mg) | 4.4 | 0.6 | 0.8 | 1.6 | 1.1 | 11.9 |

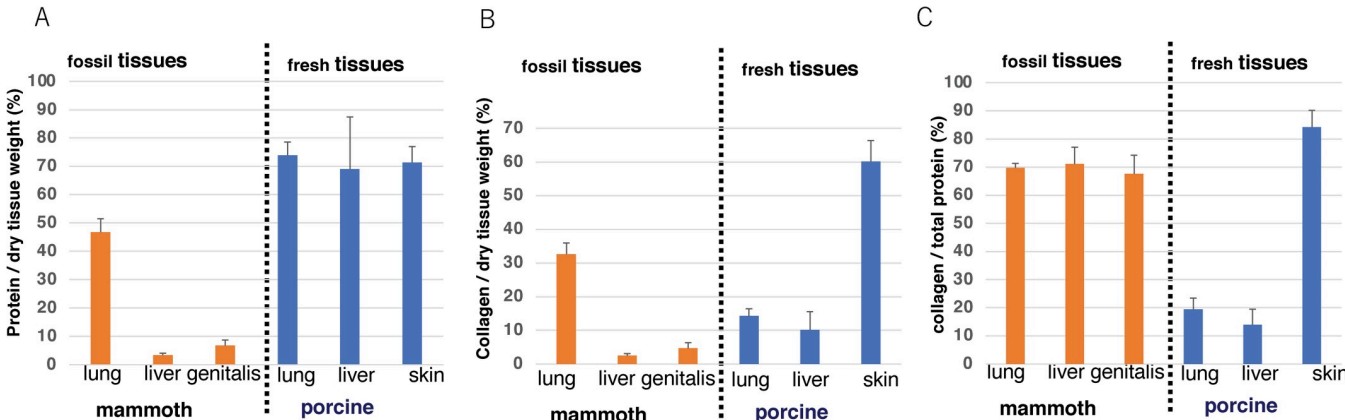

**Fig 6. Comparison of protein and collagen content in mammoth tissue and fresh tissue.** (a) Percentage of protein content against the dry weight of tissue estimated using amino acid analysis data. (b) Percentage of collagen content against the dry weight of tissue estimated using amino acid analysis data. (c) Ratio of collagen to total protein in the tissue. All the data is average of three or four independent analyses and standard deviations were indicated.

## Collagen extraction from mammoth tissue

Our results obtained by means of several different approaches suggested that collagen molecules are preserved in mammoth frozen tissue. We attempted to extract native collagen molecules from these tissues. Pepsin treatment (the classical method of collagen extraction) and alkali treatment (which is a harsh and effective method) were tried, but we were not successful in obtaining collagen molecules (Fig 8). We will attempt extraction of native collagen in future, if we can obtain samples that have not been treated (e.g. by fixation with ethanol or formalin). Once we obtain soluble collagen molecules, we can measure the melting temperature of that collagen, which may reflect the body temperature of mammoths.

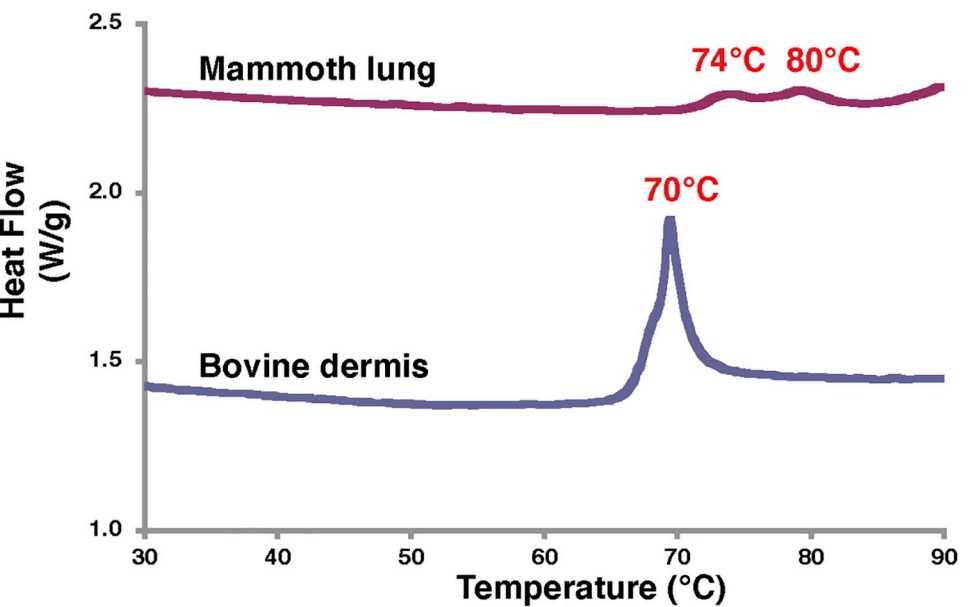

**Fig 7. DSC analysis data of lung tissue from Masha and bovine dermis.**

**Table 3. Comparison of proteins identified by LC–MS analysis in the soft tissues from Masha and tissues from the Indian elephant.**

| Sample | N | Score | Accession number | Name | Species |
|---|---|---|---|---|---|
| Mammoth lung | 1 | 35.9 | sp\|P85154\|CO1A2_MAMAE | Collagen alpha-2(I) chain | *Mammut americanum* |
| | 2 | 32.7 | sp\|P0C2W8\|CO1A1_MAMAE | Collagen alpha-1(I) chain | *Mammut americanum* |
| | 3 | 24.2 | sp\|C0HJP7\|CO1A1_TOXSP | Collagen alpha-1(I) chain (Fragments) | *Toxodon sp.* |
| | 4 | 14.6 | sp\|C0HJP8\|CO1A2_TOXSP | Collagen alpha-2(I) chain (Fragments) | *Toxodon sp.* |
| | 5 | 10.0 | sp\|Q01149\|CO1A2_MOUSE | Collagen alpha-2(I) chain | *Mus musculus* |
| | 6 | 7.8 | sp\|P04258\|CO3A1_BOVIN | Collagen alpha-1(III) chain | *Bos taurus* |
| | 7 | 2.2 | sp\|Q45XJ0\|HBD_LOXAF | Hemoglobin subunit delta | *Loxodonta africana* |
| | 8 | 2.0 | sp\|Q6DII2\|SRSF1_XENTR | Serine/arginine-rich splicing factor 1 | *Xenopus tropicalis* |
| | 9 | 0.9 | sp\|Q17RW2\|COOA1_HUMAN | Collagen alpha-1(XXIV) chain | *Homo sapiens* |
| | 10 | 0.5 | sp\|P50494\|PVDG_PLAKN | Duffy receptor gamma | *Plasmodium knowlesi* |
| Mammoth liver | 1 | 26.3 | sp\|P85154\|CO1A2_MAMAE | Collagen alpha-2(I) chain | *Mammut americanum* |
| | 2 | 21.4 | sp\|P0C2W8\|CO1A1_MAMAE | Collagen alpha-1(I) chain | *Mammut americanum* |
| | 3 | 19.1 | sp\|C0HJP3\|CO1A1_MYLDA | Collagen alpha-1(I) chain (Fragments) | *Mylodon darwinii* |
| | 4 | 15.7 | sp\|C0HJN3\|CO1A1_ORYAF | Collagen alpha-1(I) chain (Fragments) | *Orycteropus afer* |
| | 5 | 12.4 | sp\|C0HJN6\|CO1A2_HIPAM | Collagen alpha-2(I) chain (Fragments) | *Hippopotamus amphibius* |
| | 6 | 6.0 | sp\|P04258\|CO3A1_BOVIN | Collagen alpha-1(III) chain | *Bos taurus* |
| | 7 | 2.0 | sp\|Q45XJ0\|HBD_LOXAF | Hemoglobin subunit delta | *Loxodonta africana* |
| | 8 | 2.0 | sp\|Q9Z331\|K2C6B_MOUSE | Keratin, type II cytoskeletal 6B | *Mus musculus* |
| | 9 | 0.8 | sp\|Q1JUQ1\|ARADA_AZOBR | L-arabonate dehydratase | *Azospirillum brasilense* |
| | 10 | 0.7 | sp\|Q0S0N6\|YQGF_RHOJR | Putative pre-16S rRNA nuclease | *Rhodococcus jostii* |
| Elephant bone | 1 | 67.4 | sp\|P85154\|CO1A2_MAMAE | Collagen alpha-2(I) chain | *Mammut americanum* |
| | 2 | 63.9 | sp\|P0C2W8\|CO1A1_MAMAE | Collagen alpha-1(I) chain | *Mammut americanum* |
| | 3 | 32.1 | sp\|C0HJN3\|CO1A1_ORYAF | Collagen alpha-1(I) chain (Fragments) | *Orycteropus afer* |
| | 4 | 29.1 | sp\|P02465\|CO1A2_BOVIN | Collagen alpha-2(I) chain | *Bos taurus* |
| | 5 | 23.4 | sp\|C0HJP1\|CO1A1_CYCDI | Collagen alpha-1(I) chain (Fragments) | *Cyclopes didactylus* |
| | 6 | 11.9 | sp\|P02467\|CO1A2_CHICK | Collagen alpha-2(I) chain | *Gallus gallus* |
| | 7 | 6.2 | sp\|P47853\|PGS1_RAT | Biglycan | *Rattus norvegicus* |
| | 8 | 2.4 | sp\|Q9YIB4\|CO1A1_CYNPY | Collagen alpha-1(I) chain | *Cynops pyrrhogaster* |
| | 9 | 2.0 | sp\|Q6DII2\|SRSF1_XENTR | Serine/arginine-rich splicing factor 1 | *Xenopus tropicalis* |
| | 10 | 2.0 | sp\|P35441\|TSP1_MOUSE | Thrombospondin-1 | *Mus musculus* |
| | 11 | 2.0 | sp\|P22458\|VTNC_RABIT | Vitronectin | *Oryctolagus cuniculus* |
| | 12 | 1.1 | sp\|Q8PXG9\|RBL_METMA | Ribulose bisphosphate carboxylase | *Methanosarcina mazei* |
| | 13 | 1.0 | sp\|P00544\|FGR_FSVGR | Tyrosine-protein kinase transforming protein Fgr | *Feline sarcoma virus* |
| Elephant kidney | 1 | 3.8 | sp\|Q5ZLC5\|ATPB_CHICK | ATP synthase subunit beta, mitochondrial | *Gallus gallus* |
| | 2 | 2.2 | sp\|O46392\|CO1A2_CANLF | Collagen alpha-2(I) chain | *Canis lupus familiaris* |
| | 3 | 2.0 | sp\|Q5R5M8\|ECHP_PONAB | Peroxisomal bifunctional enzyme | *Pongo abelii* |
| | 4 | 2.0 | sp\|Q5R546\|ATPA_PONAB | ATP synthase subunit alpha, mitochondrial | *Pongo abelii* |
| | 5 | 1.6 | sp\|P18984\|HBB_BALAC | Hemoglobin subunit beta | *Balaenoptera acutorostrata* |
| | 6 | 1.3 | sp\|Q11DF0\|TSAD_CHESB | tRNA N6-adenosine threonylcarbamoyltransferase | *Chelativorans sp.* |
| | 7 | 1.0 | sp\|B2HRP4\|MENC_MYCMM | o-succinylbenzoate synthase | *Mycobacterium marinum* |
| | 8 | 0.8 | sp\|O18404\|HCD2_DROME | 3-hydroxyacyl-CoA dehydrogenase type-2 | *Drosophila melanogaster* |
| | 9 | 0.7 | sp\|Q9QZR9\|CO4A4_MOUSE | Collagen alpha-4(IV) chain | *Mus musculus* |
| | 10 | 0.5 | sp\|Q8TAZ6\|CKLF2_HUMAN | CKLF-like MARVEL transmembrane domain-containing protein 2 | *Homo sapiens* |

Trypsin was removed from the list.

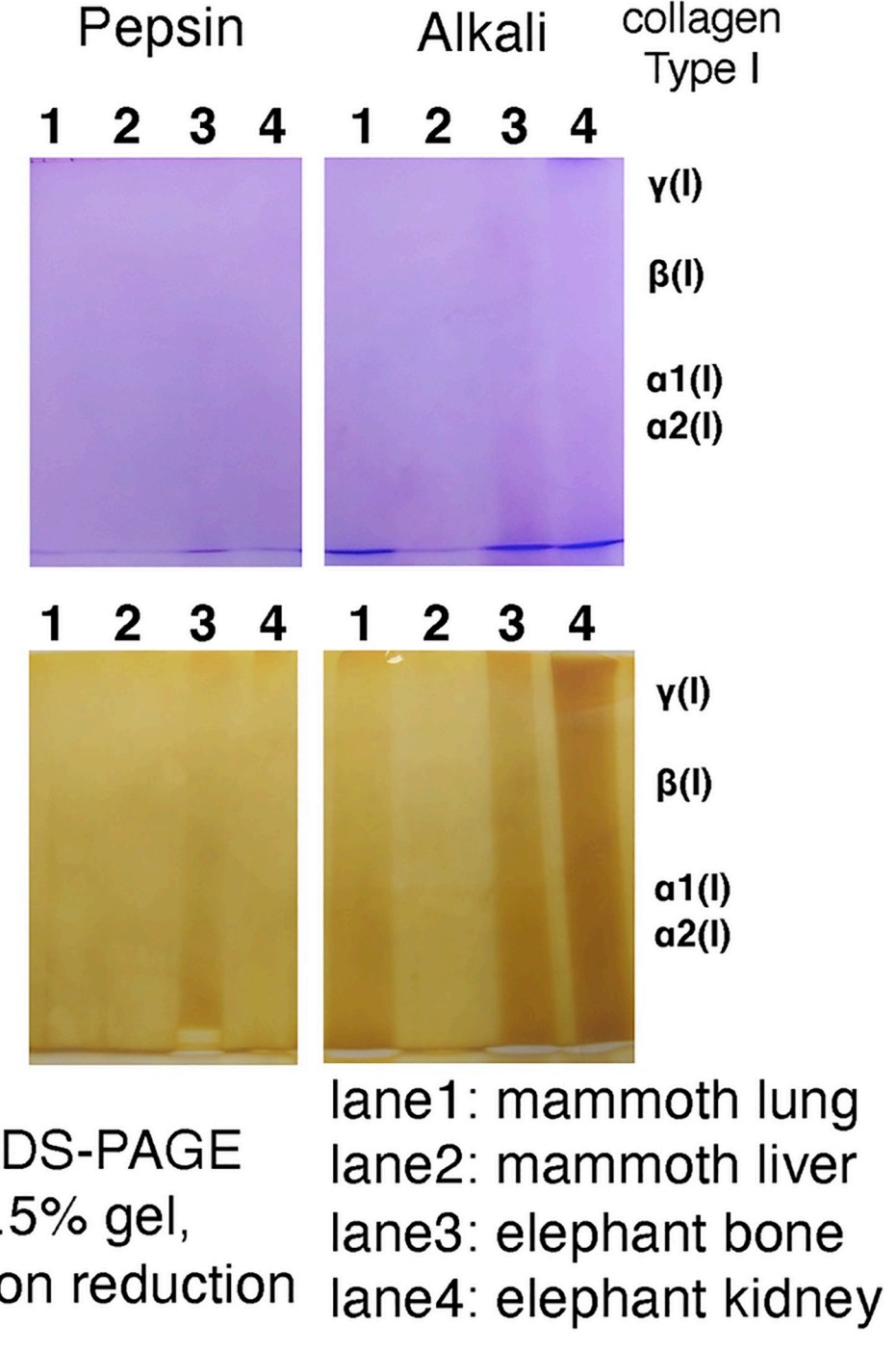

**Fig 8. Coomassie staining (upper) and silver staining (lower) of extracts of mammoth and elephant tissues after SDS-PAGE.** Right margin shows the band positions of typical type I collagen α,β and γ chains.

## Conclusions

Biochemical analysis of the famous frozen mammoths Masha and Dima has not previously been published. This is the first study of analysing collagen protein in mammoth soft tissues.

By means of histochemical observation, we found that the extracellular framework and collagen structure were well preserved in soft tissues; the presence of collagen was detected qualitatively by an immunohistochemical method. Amino acid analysis of whole tissue enabled evaluation of the quantitative profile of protein in these tissues. The amino acid composition data indicated that the majority of the preserved protein in the frozen mammoth tissue is collagen, and other types of proteins have been degraded. Collagen is a very stable molecule and abundant in animal tissue [1, 2]: here we demonstrate preservation of collagen in mammoth soft tissue for more than ten thousand years.

DSC analysis detected endothermic peaks at 74 and 80˚C, which may represent collagen structural changes. This finding may indicate that some native collagen with a triple-helical structure may be preserved in the mammoth soft tissue.

LC–MS analysis of trypsin-treated samples was conducted to identify different types of collagen. Mammoth lung and liver contain relatively high amounts of type I and type III collagen; hemoglobin was also detected. Collagen analysis from mammoth ivory or bone of mammoth has been described previously (3), but this work is the first report of collagen detection from soft mammoth tissues.

Finally, we attempted to extract collagen molecules by two different methods (pepsin digestion and alkaline treatment), but we could not succeed in solubilisation of collagen. DSC analysis suggested that the collagen in the mammoth tissue is heavily cross-linked, which may be why we could not extract collagen from frozen tissue even by alkaline treatment.

Sequencing of the nuclear genome of the woolly mammoth has been carried out [7] but post-translational protein information, for example hydroxylation of amino acids in collagen molecules, which is important for the thermal stability of proteins, cannot be obtained from the DNA sequence itself. Analysis of preserved protein will help to deepen our understanding of mammoth physiology.

## Supporting information

**S1 Raw images.**
(PDF)

## Acknowledgments

We thank Lucy Muir, PhD, from Edanz Group (https://en-author-services.edanzgroup.com/ac) for editing a draft of this manuscript.

## Author Contributions

**Conceptualization:** Shunji Hattori, Haruki Senoo.

**Data curation:** Shunji Hattori, Tomomi Kiriyama-Tanaka, Masashi Kusubata, Yuki Taga, Testuya Ebihara, Yuki Kumazawa, Katsuyuki Imai, Mitsutaka Miura, Yoshihiro Mezaki, Haruki Senoo.

**Formal analysis:** Shunji Hattori, Yuki Taga, Haruki Senoo.

**Funding acquisition:** Shunji Hattori, Yuki Taga.

**Investigation:** Shunji Hattori, Tomomi Kiriyama-Tanaka, Masashi Kusubata, Yuki Taga, Katsuyuki Imai.

**Methodology:** Shunji Hattori, Tomomi Kiriyama-Tanaka, Masashi Kusubata, Yuki Taga, Katsuyuki Imai.

**Project administration:** Shunji Hattori, Haruki Senoo.

**Resources:** Alexei Tikhonov.

**Software:** Shunji Hattori.

**Supervision:** Shunji Hattori.

**Validation:** Shunji Hattori, Haruki Senoo.

**Visualization:** Shunji Hattori.

**Writing – original draft:** Shunji Hattori.

**Writing – review & editing:** Shunji Hattori, Yuki Taga, Katsuyuki Imai, Haruki Senoo.

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
