## [Decision Letter · Decision Letter 0]

4 Oct 2021

Preservation of collagen in the soft tissues of frozen mammoths

PONE-D-21-26885

Dear Dr. Hattori,

We’re pleased to inform you that your manuscript has been judged scientifically suitable for publication and will be formally accepted for publication once it meets all outstanding technical requirements.

Kind regards,

Esmaiel Jabbari, PhD

Academic Editor

PLOS ONE

Additional Editor Comments (optional):

Reviewers' comments:

Reviewer's Responses to Questions

**Comments to the Author**

1. Is the manuscript technically sound, and do the data support the conclusions?

Reviewer #1: Yes

Reviewer #2: Yes

2. Has the statistical analysis been performed appropriately and rigorously? 

Reviewer #1: Yes

Reviewer #2: Yes

3. Have the authors made all data underlying the findings in their manuscript fully available?

Reviewer #1: Yes

Reviewer #2: Yes

4. Is the manuscript presented in an intelligible fashion and written in standard English?

Reviewer #1: Yes

Reviewer #2: Yes

5. Review Comments to the Author

Reviewer #1: This manuscript about preservation of collagen in soft tissues of frozen mammoths is interesting and well-written. I recommend accepting this manuscript. Please just correct the following typos in the final version of the manuscript.

1-Some of the figure numbers in the text are not correct. For instance, the DSC data (page 15) is presented in Figure 7, not Figure 6. Please check and correct the figure numbers in the manuscript.

2- In Table 2, please change Live to liver.

Reviewer #2: This is an excellent paper, very well written and organized. The materials and method section is written comprehensively so that the study can easily be reproduced. The article provides novel information to ECM structure of two frozen baby woolly mammoths that were buried in Siberian permafrost approximately 40,000 years ago.

6. PLOS authors have the option to publish the peer review history of their article (what does this mean?). If published, this will include your full peer review and any attached files.

Reviewer #1: No

Reviewer #2: No

---

## [Editor Report · Acceptance letter]

22 Oct 2021

PONE-D-21-26885 

Preservation of collagen in the soft tissues of frozen mammoths 

Dear Dr. Hattori:

I'm pleased to inform you that your manuscript has been deemed suitable for publication in PLOS ONE. Congratulations! Your manuscript is now with our production department. 

Kind regards, 

on behalf of

Dr. Esmaiel Jabbari 

Academic Editor

PLOS ONE